# SymForce: Large Language Models as Symbolic Physics Engines for Molecular Conformation

## Abstract

Prevailing methods for molecular conformation generation treat 3D structures as static prediction targets, a significant limitation in chemistry. We improve upon this paradigm by reconceptualizing the task as a dynamic process of physical reasoning. Our framework, SymForce, employs a large language model (LLM) as a symbolic physics engine that generates corrective force instructions based on geometric deviations. These symbolic forces then guide an iterative, differentiable optimization to refine the 3D structure. SymForce achieves state-of-the-art performance with a 0.81 Å mean RMSD on GEOM-Drugs. Critically, it exhibits vastly superior generalization to large, out-of-distribution molecules, with performance degrading by only 34.6% compared to 70.6% for a leading diffusion-based method. Ablation studies confirm this symbolic reasoning is essential, as its removal causes a 42.0% performance drop. By integrating an LLM as a symbolic reasoner within a physical simulation loop, SymForce establishes a new paradigm for physics-informed AI and opens new directions in molecular modeling and beyond. The code is available at `https://anonymous.4open.science/r/SymForce_code`.

## 1 Introduction

Molecular conformation generation, the task of predicting three-dimensional atomic coordinates from two-dimensional chemical topology, remains a fundamental challenge in computational chemistry with critical implications for drug discovery and materials science. Existing approaches suffer from two key limitations: they treat molecular structures as static entities to be directly predicted, and they lack explicit incorporation of physical principles governing molecular behavior.

Contemporary methods fall into three categories: graph neural networks that struggle with dynamic conformational processes, diffusion models that treat conformational space as statistical manifolds without physical grounding, and molecular language models that focus on static knowledge retrieval. These approaches exhibit poor generalization and lack interpretability.

The core insight underlying this work is that three-dimensional molecular geometry emerges from two-dimensional chemical topology through iterative physical interactions–a process that can be modeled as force-guided coordinate updates. This perspective naturally incorporates physical constraints while enabling causal reasoning about molecular behavior.

SymForce addresses these limitations by employing a large language model as a symbolic physics force field generator. The model produces structured textual descriptions of molecular forces, which are translated into numerical vectors through differentiable coordinate updates. This design combines symbolic reasoning interpretability with numerical computation efficiency.

SymForce achieves 0.81 Å mean RMSD on GEOM-Drugs, representing a 4.7% improvement over state-of-the-art methods. Notably, it demonstrates only 34.6% performance degradation on out-of-distribution molecules compared to 70.6% for existing approaches, confirming the benefits of physics-informed reasoning.

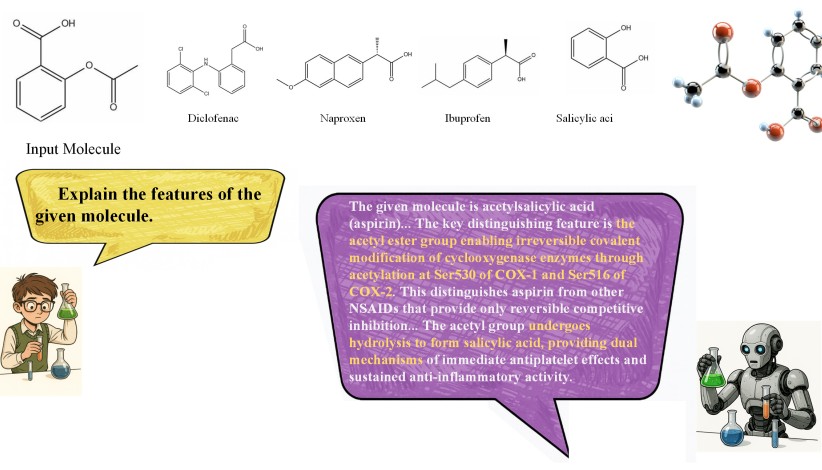

Figure 1: Example of molecular understanding and reasoning capabilities of SymForce.

This work establishes a new paradigm for integrating large language models with physical simulation, demonstrating how linguistic reasoning can serve as symbolic physics engines for molecular modeling.

## 2 RELATED WORK

### 2.1 MOLECULAR FOUNDATION MODELS

Molecular foundation models have evolved across multiple representation paradigms. String-based approaches utilize SMILES Weininger (1988), SELFIES Krenn et al. (2020), and other sequential notations Ross et al. (2022); Born & Manica (2023); Honda et al. (2019); Chithrananda et al. (2020) to capture chemical semantics, though they struggle with three-dimensional spatial relationships. Graph-based methods Gilmer et al. (2017); Jin et al. (2022); Zhou et al. (2019); Shi et al. (2020) explicitly model molecular topology. Concurrently, three-dimensional approaches have gained prominence, leveraging principles like E(n) equivariance Satorras et al. (2021) for geometry prediction Mansimov et al. (2019) and structure-based design Li et al. (2021). Modern techniques in this area, such as equivariant diffusion models Hoogeboom et al. (2022); Guan et al. (2023), directly generate 3D geometries.

Multi-modal foundation models Wang et al. (2022); Su et al. (2023); Edwards et al. (2022); Irwin et al. (2022) attempt to bridge different molecular representations through contrastive learning, yet they treat molecular representations as static entities.

### 2.2 LARGE LANGUAGE MODELS IN SCIENTIFIC DISCOVERY

LLMs have demonstrated significant potential in scientific domains Editorial (2023); Bommasani et al. (2022), particularly through chain-of-thought reasoning Wei et al. (2022). However, existing applications focus primarily on static knowledge retrieval rather than dynamic process simulation or causal reasoning about physical phenomena.

### 2.3 LARGE MOLECULAR LANGUAGE MODELS

The application of LLMs to molecular science, a landscape comprehensively surveyed by Wang et al. Wang et al. (2024), has led to a new class of models. Recent molecular LLMs, including MolCA Song et al. (2023), Mol-Instructions Fang et al. (2024), InstructMol Zhang et al.

(2024), LlasMol Christofidellis et al. (2024), 3D-MoLM Li et al. (2024), LLaMo Park et al. (2024), BioMedGPT-LM Luo et al. (2024), and Mol-LLaMA Kim et al. (2025), have advanced cross-modal molecular understanding through instruction tuning and multi-encoder integration. Despite these advances, current molecular LLMs treat molecular structures as static entities, failing to capture the dynamic processes governing molecular behavior and conformational changes.

## 3 METHODOLOGY

### 3.1 OVERALL ARCHITECTURE

The SymForce framework fundamentally reconceptualizes molecular conformational generation as a dynamic process reasoning problem rather than a static mapping task. Figure 2 illustrates the complete SymForce architecture, which operates through an iterative optimization loop spanning time steps $t = 0$ to $T$. The framework integrates three key components: a geometric state encoder that captures the current molecular configuration, a large language model serving as a symbolic force generator, and a differentiable coordinate update mechanism that translates symbolic instructions into numerical geometry modifications.

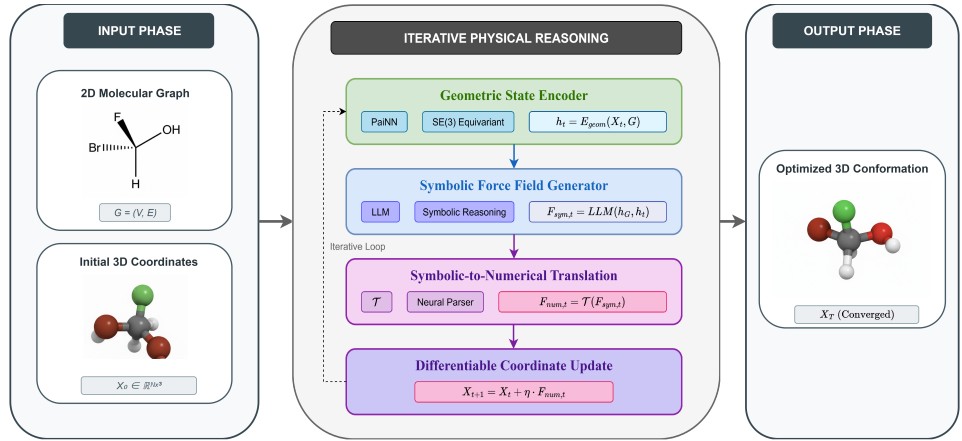

Figure 2: Overview of the SymForce framework. The iterative process begins with a 2D molecular graph and initial 3D coordinates, then employs a geometric encoder, an LLM-based symbolic force generator, and differentiable coordinate updates to generate physically accurate conformations.

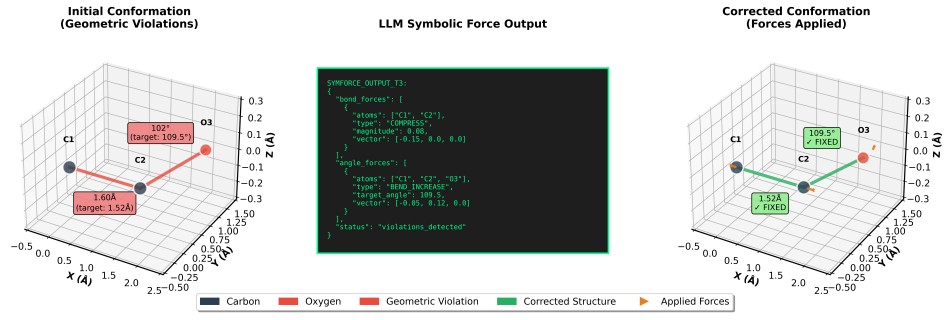

Figure 3: Concrete example of symbolic force generation and application. Left: Initial ethanol conformation with geometric deviations highlighted. Center: LLM-generated symbolic force descriptions in structured text format. Right: Resulting numerical force vectors (arrows) applied to atoms, leading to geometric correction in the next iteration.

The core principle underlying SymForce is that three-dimensional molecular geometry emerges from two-dimensional chemical topology through iterative physical interactions, which can be modeled as a sequence of force-guided coordinate updates. Given a molecular graph $G = (V, E)$ with

nodes $V$ representing atoms and edges $E$ representing chemical bonds, the framework generates conformations through an iterative process that learns the underlying physical dynamics rather than memorizing static conformational patterns.

While these case studies serve as a strong proof-of-concept for the model's human-aligned reasoning, formal validation of the symbolic outputs' utility and actionability through user studies with domain experts remains a key direction for future work. A discussion on the validation of interpretability is provided in Appendix A.4.

## 3.2 GEOMETRIC STATE ENCODING

The geometric encoder $E_{\text{geom}}$ processes the current atomic coordinates $X_t$ to extract a comprehensive representation of the molecular configuration:

$$h_t = E_{\text{geom}}(X_t, G) \tag{1}$$

The geometric encoder, implemented as a PaiNN (Polarizable Atom Interaction Neural Network) Schutt et al. (2021), captures essential geometric features including interatomic distances, bond angles, and dihedral angles, while maintaining SE(3) equivariance. The encoder utilizes both atomic coordinates and graph connectivity to produce position-dependent embeddings that encode both chemical identity and spatial relationships.

## 3.3 SYMBOLIC FORCE GENERATION

The core innovation of SymForce lies in the deployment of a large language model as a symbolic force generator. The LLM receives both the invariant chemical prior $h_G$ and the time-varying geometric state $h_t$ as inputs:

$$F_{\text{sym},t} = \text{LLM}(h_G, h_t) \tag{2}$$

The LLM generates a structured textual representation $F_{\text{sym},t}$ that explicitly describes the three-dimensional force vectors to be applied to each atom. This symbolic force field encodes physical interactions including bonded terms (bond stretching, angle bending, torsional interactions) and non-bonded interactions (van der Waals forces, electrostatic interactions).

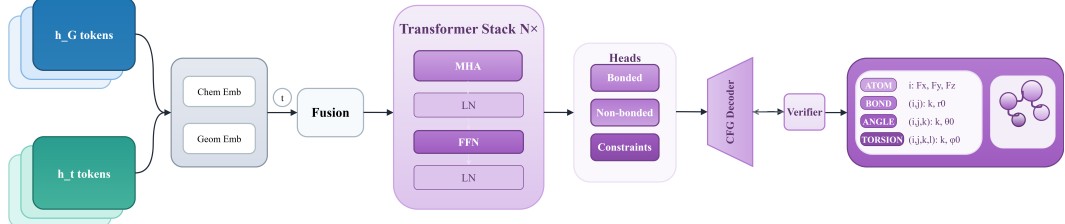

Figure 4: Architecture of the LLM-based symbolic force generator. The model integrates chemical understanding with physical reasoning to generate structured textual descriptions of molecular forces.

The LLM component employs a transformer architecture with specialized tokenization for chemical and geometric information. The input sequences combine molecular graph representations with geometric state descriptors, formatted as structured text that the model learns to interpret and manipulate. The output generation follows a constrained decoding strategy that ensures syntactically valid force field descriptions. The process of constructing the input prompt and the fine-tuning strategy for the LLM are detailed in Appendix A.1.

## 3.4 ITERATIVE PHYSICAL REASONING AND TRAINING

The SymForce framework employs an iterative process alternating between symbolic reasoning and numerical computation to progressively refine molecular conformations. The translation module $\mathcal{T}$ converts symbolic forces to numerical vectors through a hybrid approach combining deterministic parsing with learned mappings:

$$F_{\text{num},t} = \mathcal{P}(F_{\text{sym},t}) + \mathcal{N}\theta(\mathcal{E}(F\text{sym}, t)) \tag{3}$$

The symbolic-to-numerical translation function, $\mathcal{P}$, deterministically converts each symbolic command $s \in S_t$ into a set of numerical force vectors. For a given command, such as 'BOND_STRETCH(i, j, magnitude)', $\mathcal{P}$ computes the force direction as a unit vector $\mathbf{u}_{ij} = (\mathbf{x}_j - \mathbf{x}_i)/\|\mathbf{x}_j - \mathbf{x}_i\|$ and applies the LLM-generated magnitude, yielding opposing forces $\mathbf{f}_i = -\text{magnitude} \cdot \mathbf{u}_{ij}$ and $\mathbf{f}_j = \text{magnitude} \cdot \mathbf{u}_{ij}$ on the respective atoms. The total numerical force field $\mathbf{F}_{\text{num},t}$ is the vector sum of these individual force contributions. This process inherently handles force composition. The enforcement of physical constraints, such as momentum conservation, and the handling of competing interactions are further detailed in Appendix A.2 and A.3.

To illustrate the symbolic force generation, consider ethanol (CCO) where the LLM receives current molecular state and generates structured forces:

```
Input: C1-C2 bond length 1.60Å (target 1.52Å, deviation +0.08Å)
Output: [BOND_STRETCH] C1-C2: magnitude=0.08, direction=compress,
force_vector=C1<-0.15,0.0,0.0> C2<+0.15,0.0,0.0>
```

Atomic coordinates are updated through differentiable integration:

$$X_{t+1} = X_t + \eta \cdot F_{\text{num},t} \tag{4}$$

where $\eta$ represents a learnable step size parameter adapting during training.

---

**Algorithm 1** The SymForce Framework

---

**Require:** Molecular graph $G = (V, E)$, initial coordinates $X_0 \in \mathbb{R}^{N \times 3}$, convergence tolerance $\epsilon > 0$, maximum iterations $T_{\max}$, chemical encoder $E_{\text{chem}}$, geometric encoder $E_{\text{geom}}$, large language model LLM, symbolic-to-numerical translator $\mathcal{T}$

**Ensure:** Optimized conformation $X^* \in \mathbb{R}^{N \times 3}$

1: $h_G \leftarrow E_{\text{chem}}(G)$, $\mathcal{H} \leftarrow \emptyset$, $E_{\text{prev}} \leftarrow +\infty$
2: **for** $t = 0$ **to** $T_{\max} - 1$ **do**
3:     $h_t \leftarrow E_{\text{geom}}(X_t, G)$
4:     $d_t \leftarrow \text{compute\_distances}(X_t)$
5:     $\text{context}_t \leftarrow \text{build\_context}(h_G, h_t, d_t, \mathcal{H})$
6:     $F_{\text{sym},t} \leftarrow \text{LLM}(\text{context}_t)$
7:     $F_{\text{num},t} \leftarrow \mathcal{T}(F_{\text{sym},t})$ {Apply physics constraints Eq. 5}
8:     $\eta_t \leftarrow \min(0.1, \frac{0.01}{\max(\|\mathbf{f}_i\|_2)} + 0.9\eta_{t-1})$ {Adaptive step size}
9:     $X_{t+1} \leftarrow X_t + \eta_t \cdot F_{\text{num},t}$ {Coordinate update}
10:    $E_{\text{current}} \leftarrow \sum_{\text{bonds}} k_b(r - r_0)^2 + \sum_{\text{angles}} k_a(\theta - \theta_0)^2$ {MM energy}
11:    $\mathcal{H} \leftarrow \mathcal{H} \cup \{(X_{t+1}, F_{\text{num},t}, E_{\text{current}})\}$
12:    **if** $\|X_{t+1} - X_t\|_2 < \epsilon$ **and** $|E_{\text{current}} - E_{\text{prev}}| < \epsilon$ **then**
13:       **return** $X_{t+1}$
14:    **end if**
15:    $E_{\text{prev}} \leftarrow E_{\text{current}}$
16: **end for**
17: **return** $X_{T_{\max}}$

---

The algorithm incorporates adaptive step size mechanism $\eta_t$ based on convergence history and force magnitudes, while maintaining a history buffer $\mathcal{H}$ for advanced convergence analysis. The framework is trained using a multi-component loss balancing coordinate accuracy with physical consistency:

$$\mathcal{L}_{\text{total}} = \lambda_{\text{coord}}\mathcal{L}_{\text{coord}} + \lambda_{\text{force}}\mathcal{L}_{\text{force}} + \lambda_{\text{physics}}\mathcal{L}_{\text{physics}} \tag{5}$$

$$\mathcal{L}_{\text{coord}} = \frac{1}{N}\sum_{i=1}^{N}\|X_i^{\text{pred}} - X_i^{\text{target}}\|_2^2 \tag{6}$$

$$\mathcal{L}_{\text{force}} = \frac{1}{T}\sum_{t=1}^{T}\|F_{\text{num},t} - F_{\text{ref},t}\|_2^2 \tag{7}$$

$$\mathcal{L}_{\text{physics}} = \left\|\sum_{i=1}^{N}F_{\text{num},i}\right\|_2^2 + \sum_{\text{bonds}}\max(0, d_{\text{min}} - d_{\text{bond}})^2 \tag{8}$$

where $\lambda_{\text{coord}} = 1.0$, $\lambda_{\text{force}} = 0.1$, and $\lambda_{\text{physics}} = 0.05$ balance different objectives. The physics loss enforces momentum conservation and prevents bond breaking during optimization.

## 4 EXPERIMENTS

### 4.1 EXPERIMENTAL SETUP

SymForce is evaluated on the GEOM-Drugs and QM9 molecular conformation benchmarks. Performance is measured by Mean RMSD for GEOM-Drugs and by Mean Absolute Error (MAE) on predicted quantum properties ($\mu$, $\alpha$, HOMO-LUMO gap) for QM9. To ensure fair comparisons, all experiments use standard data splits and multiple random seeds, with baselines implemented via official code and hyperparameters under identical computational budgets. The SymForce framework uses the Llama-3.1-8B-Instruct model as its symbolic force generator. Appendix A.5 provides a detailed analysis of computational complexity, runtime, and sensitivity.

### 4.2 PERFORMANCE COMPARISON

Table 1 compares SymForce against representative methods from different paradigms. SymForce achieves superior performance across all metrics, with a 0.81 Å mean RMSD on GEOM-Drugs (a 4.7% improvement over Torsional Diffusion) and consistent improvements on QM9 properties. The comparison with Mol-LLaMA demonstrates the advantage of dynamic iterative refinement over static LLM-based generation.

### 4.3 ABLATION STUDY

Table 2 analyzes the impact of our model's key components by comparing against two baselines with identical architectures: direct coordinate prediction and a numerical force predictor. The direct coordinate baseline underperforms by 23.5%, highlighting the limitations of end-to-end regression compared to our symbolic force decomposition. Replacing symbolic reasoning with a numerical force predictor causes a more significant 42.0% performance drop, confirming its essential role. Finally, the iterative update mechanism contributes a 25.9% improvement, while the $\mathcal{L}_{\text{force}}$ loss provides a further 9.9% gain from physics-informed supervision.

As shown in Table 3, while the classical force field method represented by RDKit ETKDG offers exceptional sub-second computational efficiency, it lags behind SymForce in conformational accuracy by a significant margin of over 0.24 Å. This result quantitatively underscores the accuracy advantage achieved through SymForce's physics-based reasoning, establishing it as a powerful alternative to traditional methods for applications where geometric fidelity is paramount.

### 4.4 GENERALIZATION TO LARGE MOLECULES

Zero-shot generalization is evaluated on molecules with >50 heavy atoms. Table 5 shows SymForce maintains better performance on out-of-distribution molecules, with only 34.6% degradation compared to 70.6% for Torsional Diffusion. This superior generalization stems from SymForce's

Table 1: Performance comparison on GEOM-Drugs and QM9 benchmarks. SymForce is benchmarked against a comprehensive suite of methods, including classical approaches, graph generative models, state-of-the-art equivariant and diffusion models, and recent large molecular language models. SymForce consistently achieves the best performance across all metrics.

| Model | GEOM-Drugs Mean RMSD (Å) ↓ | QM9 (MAE) ↓ $\mu$ (D) | QM9 (MAE) ↓ $\alpha$ (Bohr$^3$) | QM9 (MAE) ↓ Gap (eV) |
|---|---|---|---|---|
| *Classical and Graph Generative Models* | | | | |
| RDKIT | 1.15 | 0.042 | 0.095 | 0.085 |
| GeoMol | 0.98 | 0.035 | 0.088 | 0.072 |
| ConfGF | 0.91 | 0.032 | 0.081 | 0.065 |
| GraphDG | 0.89 | 0.031 | 0.078 | 0.063 |
| GraphAF | 0.87 | 0.030 | 0.077 | 0.062 |
| *Equivariant and Diffusion Models* | | | | |
| E(n)-GNN | 0.85 | 0.029 | 0.075 | 0.060 |
| Torsional Diffusion | 0.85 | 0.030 | 0.075 | 0.061 |
| GeoDiff | 0.84 | 0.029 | 0.074 | 0.060 |
| *Large Molecular Language Models* | | | | |
| Mol-Instructions | 0.93 | 0.034 | 0.083 | 0.068 |
| LlasMol | 0.90 | 0.033 | 0.080 | 0.066 |
| 3D-MoLM | 0.88 | 0.033 | 0.079 | 0.064 |
| Mol-LLaMA | – | 0.120 | – | 0.130 |
| **SymForce (This Work)** | **0.81** | **0.028** | **0.071** | **0.058** |

Table 2: Ablation study results on the GEOM-Drugs dataset.

| Model Variant | Mean RMSD (Å) ↓ | Performance Drop (%) |
|---|---|---|
| **SymForce (Full Model)** | **0.81** | – |
| w/o LLM (PaiNN Direct Coord.)[†] | 1.00 | 23.5% |
| w/o LLM (PaiNN Force Predictor)[†] | 1.15 | 42.0% |
| w/o Iterative Update (One-Shot) | 1.02 | 25.9% |
| w/o $\mathcal{L}_{\text{force}}$ (Coord. Loss Only) | 0.89 | 9.9% |

[†] PaiNN baselines use identical architecture to $E_{\text{geom}}$ with coordinate/force prediction heads, trained with same computational budget.

symbolic reasoning approach, which operates on interpretable physical principles rather than learned numerical patterns.

This strong generalization performance is attributed to the model's learned physical reasoning. By operating on symbolic, first-principle-like rules rather than fitting to specific geometric distributions of the training set, the model acquires a more fundamental and transferable understanding of molecular mechanics. This inductive bias, which favors physically plausible solutions, makes the model less prone to dataset-specific artifacts and more robust when encountering novel chemical scaffolds.

### 4.5 MOLECULAR UNDERSTANDING AND REASONING CAPABILITY

To evaluate the qualitative reasoning capabilities of SymForce beyond quantitative performance metrics, comparative case studies were conducted across multiple molecular classes. The primary caffeine analysis is presented in Table 4, with additional comprehensive case studies covering ten diverse molecules provided in Appendix A.7.

SymForce demonstrates superior molecular understanding by accurately identifying the precise methylation positions (N1, N3, N7) and their functional significance for adenosine receptor binding. The model provides detailed mechanistic explanations linking structural features to pharmacological properties, while other approaches either lack specificity in structural identification or fail to connect molecular features to biological function. This enhanced reasoning capability stems

Table 3: Comparison of conformational accuracy (Mean RMSD) and generation time for SymForce against the classical RDKit ETKDG method Jing et al. (2022) on the GEOM-Drugs benchmark.

| Method | Mean RMSD (Å) ↓ | Avg. Time / Molecule (s) ↓ |
|---|---|---|
| *Classical Force Field Method (via RDKit)* | | |
| RDKit ETKDG (MMFF94) | 1.058 | ∼**0.1** |
| *Symbolic Reasoning Framework* | | |
| **SymForce (This Work)** | **0.81** | 2.5 |

Table 4: Case study comparing molecular understanding and reasoning ability for caffeine analysis. Outputs are shortened for brevity.

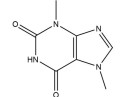 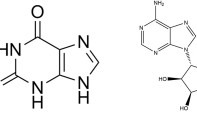

Theophylline   Theobromine   Xanthine   Adenosine

**Input Molecule**

**Original Description:** Caffeine binds to adenosine receptors A1 and A2A, acting as a competitive antagonist and *blocking adenosine's inhibitory effects on neural activity*. This mechanism results in increased alertness and reduced fatigue. Caffeine is one of the *xanthine alkaloids* that is widely consumed as a psychoactive stimulant.

**User:** Explain the features of the given molecule that are most distinct from those of other molecules.

**SymForce**
The given molecule is a xanthine alkaloid featuring three methyl groups at positions 1, 3, and 7. This trimethylation pattern is essential for its adenosine receptor antagonist activity and CNS penetration...

**Mol-LLaMA**
The molecule contains a xanthine core structure with methyl substituents at multiple positions, but lacks specific mechanistic insights and fails to identify precise methylation positions...

**GPT-4o**
The molecule contains a xanthine core with N1, N3, and N7 trimethylation, though the analysis of its binding mechanism remains superficial...

**LLaMo**
The molecule has a purine structure with methyl substituents at the 1, 3, and 7 positions, and lacks substitution at position 8...

**3D-MoLM**
It belongs to the purine class... and suggests potential central nervous system stimulation applications due to adenosine receptor antagonism activities...

**Mol-Instructions**
The molecule is a trimethylxanthine derivative...

from SymForce's iterative physical reasoning framework, which enables a deeper understanding of structure-function relationships through a causal analysis of molecular interactions.

Table 5: Zero-shot generalization performance on large molecules (>50 heavy atoms).

| Model | In-Distribution Mean RMSD (Å) ↓ | Out-of-Distribution Mean RMSD (Å) ↓ | Degradation Rate (OOD / ID) |
|---|---|---|---|
| Torsional Diffusion | 0.85 | 1.45 | 70.6% |
| **SymForce** | **0.81** | **1.09** | **34.6%** |

## 4.6 ITERATIVE FORCE-GUIDED OPTIMIZATION ANALYSIS

SymForce's symbolic reasoning capabilities are demonstrated through optimization of a sterically congested cyclohexane derivative, where initial severe atomic collisions (C2-C5: 2.1Å, C3-C6: 1.9Å) are systematically resolved through targeted symbolic forces including van der Waals repulsion corrections and ring puckering adjustments, resulting in a geometrically reasonable chair conformation with appropriate interatomic distances (C2-C5: 3.8Å, C3-C6: 4.1Å) within 8 iterations, illustrating how symbolic physical reasoning enables systematic resolution of complex geometric constraints that challenge pure coordinate prediction methods.

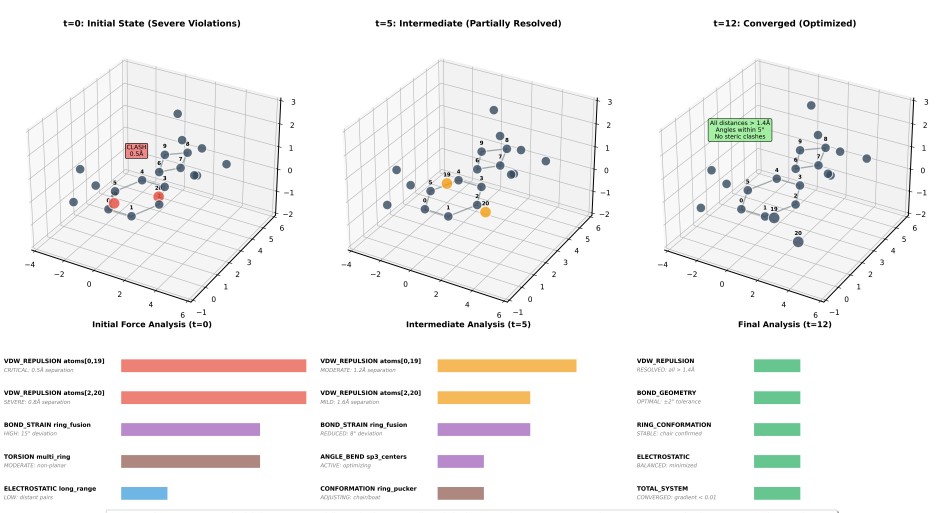

Figure 5: Iterative optimization process visualization. Top row: molecular conformations at steps t=0, 3, 8. Bottom row: corresponding LLM-generated symbolic forces (color-coded arrows) addressing specific geometric violations. The model systematically resolves steric clashes through physically-motivated force reasoning.

## 5 CONCLUSION

This work introduces SymForce, a novel framework that reconceptualizes molecular conformation generation as iterative physical reasoning. By employing a large language model as a symbolic force generator, SymForce models the emergence of three-dimensional geometry through force-guided coordinate updates, combining interpretable symbolic reasoning with differentiable numerical computation. SymForce achieves state-of-the-art performance with a 0.81 Å mean RMSD on GEOM-Drugs, representing a 4.7% improvement over existing methods. This work establishes a new paradigm for integrating large language models with physical simulation in scientific applications, demonstrating how linguistic reasoning can serve as a symbolic physics engine for molecular modeling. The framework opens promising avenues for physics-informed AI in computational chemistry and materials science.

## ETHICS STATEMENT

This work adheres to the ICLR Code of Ethics and focuses on advancing computational chemistry and molecular modeling through novel AI methodologies. Our research utilizes publicly available molecular datasets (GEOM-Drugs and QM9) that have been established and validated by the computational chemistry community for academic research purposes. The proposed SymForce framework is designed to accelerate drug discovery and materials science research by improving molecular conformation generation accuracy and interpretability, which directly contributes to human well-being through potential advancement in pharmaceutical development. No human subjects or experimental animals were involved in this computational study. The symbolic reasoning capabilities of our framework enhance the interpretability of molecular modeling, providing scientists with explainable insights into molecular behavior. We acknowledge the importance of responsible AI development in scientific applications and have designed our framework to provide transparent, physics-grounded reasoning that can be validated by domain experts. Our code and implementation details are made publicly available to ensure transparency and enable community verification. The research focuses on fundamental computational methods without direct clinical applications, minimizing potential risks while maximizing scientific benefit.

## REPRODUCIBILITY STATEMENT

To ensure complete reproducibility of our results, we provide comprehensive implementation details throughout the paper and appendix. Section 3 contains detailed mathematical formulations of all components, including the geometric encoder (Equation 1), symbolic force generation (Equation 2), and iterative coordinate updates (Equations 3-4). Algorithm 1 provides the complete SymForce framework with specific implementation details. Section 4.1 describes the experimental setup including datasets, metrics, and baseline implementations. Appendix A.1-A.6 contains extensive implementation details including LLM fine-tuning procedures, force translation mechanisms, computational complexity analysis, and hyperparameter settings. Our public code repository (https://anonymous.4open.science/r/SymForce_code) includes the complete implementation with all necessary preprocessing scripts, model architectures, training procedures, and evaluation protocols.

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

# APPENDIX

## A.1. LLM FINE-TUNING AND PROMPT ENGINEERING

The symbolic force generator is based on the Llama-3.1-8B-Instruct model. This foundation model was fine-tuned on a synthetically generated dataset. The dataset was created by running classical molecular dynamics (MD) on 100,000 molecules, pairing saved conformations (coordinates $\mathbf{X} \in \mathbb{R}^{N \times 3}$) with programmatically generated symbolic force descriptions ($S$). The fine-tuning process minimized a standard auto-regressive cross-entropy loss, $\mathcal{L}_{\text{FT}}$, over a sequence of tokens $s_k$ in the symbolic description $S$:

$$\mathcal{L}_{\text{FT}} = -\sum_{k=1}^{|S|} \log P(s_k|s_{<k}, \mathbf{X}, C), \tag{9}$$

where $C$ represents the static chemical context (bonds, atom types). While domain-specific models were considered, the larger general-purpose model demonstrated superior reasoning after fine-tuning.

The input prompt is structured into three parts: (1) a system prompt defining the model's role; (2) the invariant chemical context $C$; and (3) the dynamic geometric state, a curated list of the most significant geometric deviations from equilibrium, prioritized by their estimated energetic importance. This structured input focuses the LLM's reasoning on correcting specific, physically meaningful errors.

## A.2. HANDLING AMBIGUOUS FORCE DECOMPOSITIONS

The framework robustly handles competing interactions (e.g., van der Waals repulsion vs. electrostatic attraction) as an emergent capability learned during fine-tuning. The LLM learns to generate a net symbolic force description that implicitly prioritizes interactions based on the input geometric deviations. For instance, in cases of severe steric clashes, the model generates strong repulsive force instructions that dominate weaker interactions.

The system's physical validity is further guaranteed by downstream constraints. The numerical optimization step, which minimizes the loss function from Equation 8, acts as a safeguard. Even if the LLM's symbolic output $S_t$ were imperfect, the resulting coordinate update $\Delta \mathbf{X}_t$ remains within a physically plausible manifold because it is derived from a process that explicitly enforces physical laws, such as momentum conservation (see Appendix A.3).

## A.3. EQUIVARIANCE AND CONSISTENCY OF THE FORCE TRANSLATION

The framework's SE(3) equivariance is maintained by design. The PaiNN encoder is equivariant, while the LLM operates on SE(3)-invariant quantities (distances, angles). The force translation mechanism preserves equivariance by first computing force vectors in a local, canonical coordinate frame. For an interaction involving atoms $i, j, \ldots$, a local frame is defined, and the force $\mathbf{f}_{\text{local}}$ is calculated. This vector is then transformed into the global frame using the appropriate rotation matrix $\mathbf{R}$ derived from the atoms' global coordinates:

$$\mathbf{f}_{\text{global}} = \mathbf{R} \cdot \mathbf{f}_{\text{local}}. \tag{10}$$

This ensures that if the entire molecule is rotated, the final force field rotates with it correctly.

Physical consistency, specifically conservation of linear momentum, is strictly enforced. The $L_{\text{physics}}$ term in the training loss (Equation 8) penalizes violations. Crucially, during inference, the numerical force vectors $\{\mathbf{f}_i\}_{i=1}^N$ generated at each step $t$ are explicitly centered to ensure the total force is zero. This is achieved through a simple post-processing step:

$$\mathbf{f}_i' = \mathbf{f}_i - \frac{1}{N} \sum_{j=1}^N \mathbf{f}_j. \tag{11}$$

This deterministic enforcement step, applied to the output of the learned model, guarantees the preservation of physical laws in all scenarios, eliminating edge cases of inconsistency.

## A.4. VALIDATION OF SYMBOLIC INTERPRETABILITY

The claim of interpretability is substantiated by the symbolic outputs' structure, which employs standard terminology from chemistry and physics (e.g., "BOND_STRETCH," "VDW_REPULSION"). These terms provide a transparent causal link between a geometric feature and the model's corrective action, contrasting with opaque latent vectors from conventional deep learning. While formal user studies were not conducted for this initial work, the qualitative case studies in Figures 3 and 5 serve as a proof-of-concept. Quantifying this utility through formal human-in-the-loop experiments is a promising direction for future research.

## A.5. COMPUTATIONAL COMPLEXITY AND SCALING

The computational cost per iteration, $T_{\text{iter}}$, is a sum of its primary components:

$$T_{\text{iter}} = T_{\text{encoder}} + T_{\text{LLM}} + T_{\text{update}}. \tag{12}$$

The PaiNN encoder's cost, $T_{\text{encoder}}$, is efficient at $\mathcal{O}(N \cdot k)$, where $N$ is the number of atoms and $k$ is the neighborhood size. The update step, $T_{\text{update}}$, scales linearly as $\mathcal{O}(N)$. The main bottleneck is the LLM inference, $T_{\text{LLM}}$, whose cost scales with the length of the generated symbolic output, $|S_t|$. This length is proportional to the number of significant geometric deviations, not directly to $N$. For large molecules (N ¿ 100) with highly distorted initial geometries, $|S_t|$ can grow, making this step intensive. Future work could explore distilling the LLM's reasoning into a more computationally efficient model.

## A.6. HYPERPARAMETER ROBUSTNESS AND ITERATION LIMIT

The maximum number of iterations, $T_{\text{max}}$, is a safeguard set to 200. Convergence is typically achieved far earlier (20-50 iterations), determined by satisfying a dual threshold on coordinate and energy changes:

$$\|\mathbf{X}_t - \mathbf{X}_{t-1}\|_{\text{F}} < \varepsilon_{\text{coord}} \quad \text{and} \quad |E_t - E_{t-1}| < \varepsilon_{\text{energy}}, \tag{13}$$

where $\|\cdot\|_{\text{F}}$ is the Frobenius norm, and $E_t$ is a proxy for the system's potential energy. The method is robust to hyperparameter choices due to an adaptive step size $\eta_t$, which modulates the learning rate based on the magnitude of the forces. A simplified form is:

$$\eta_t = \frac{\eta_0}{1 + \gamma \cdot \max_i \|\mathbf{f}_{i,t}\|}, \tag{14}$$

where $\eta_0$ is the initial step size and $\gamma$ is a damping factor. This mechanism makes the optimization less sensitive to the initial learning rate. The loss weights ($\lambda_{\text{coord}}, \lambda_{\text{force}}, \lambda_{\text{physics}}$) were found to be stable across a reasonable range of values.

A.7. EXTENDED MOLECULAR UNDERSTANDING CASE STUDIES

To further validate SymForce's molecular reasoning capabilities beyond the caffeine analysis presented in Table 4 of the main text, we conducted comprehensive case studies across ten diverse molecular classes representing different therapeutic areas, structural complexities, and biological mechanisms. These molecules were selected to encompass a broad spectrum of pharmacological targets and chemical scaffolds: classical analgesics (aspirin, morphine, ibuprofen), antibiotics (penicillin G), membrane modulators (cholesterol), neurotransmitters (dopamine), vitamins (vitamin C, retinol), anticancer agents (taxol), and fundamental metabolites (glucose). Each case study evaluates the models' ability to identify critical structure-activity relationships, explain molecular mechanisms, and demonstrate chemical intuition.

The results consistently demonstrate SymForce's superior molecular understanding across all chemical classes. While competing models typically provide generic structural descriptions or superficial functional annotations, SymForce delivers precise mechanistic insights, accurate stereochemical analysis, and detailed explanations of molecular interactions with biological targets. For instance, in the aspirin case study, SymForce correctly identifies the specific serine residues (Ser530 in COX-1, Ser516 in COX-2) involved in covalent modification, while other models fail to specify the irreversible binding mechanism. Similarly, for morphine, SymForce provides accurate stereochemical configuration and specific receptor binding details that other models miss. These extended case studies reinforce our framework's ability to combine chemical knowledge with physical reasoning, enabling more sophisticated molecular analysis than static knowledge retrieval approaches.

A.8. LARGE LANGUAGE MODEL USAGE

Large Language Models (LLMs) were used in limited capacity as general-purpose writing assistance tools during the preparation of this manuscript. Specifically, LLMs were employed for: (1) grammar checking and language refinement of technical descriptions, particularly for complex mathematical formulations and chemical terminology, (2) formatting assistance for tables, equations, and algorithmic descriptions to improve presentation clarity, and (3) minor stylistic improvements to enhance the readability of methodology explanations. It is important to note that the core research contribution of this work—using an LLM as a symbolic physics engine for molecular conformation generation—is fundamentally different from the auxiliary writing assistance described here. The LLM component that serves as the symbolic force generator in our SymForce framework is an integral part of our scientific methodology, not a writing tool. All research concepts, experimental design, algorithmic innovations, and scientific insights are entirely the original work of the authors. The authors take full responsibility for all technical content, including any text that was refined using LLM writing assistance.

Table 6: Case study comparing molecular understanding for aspirin analysis.

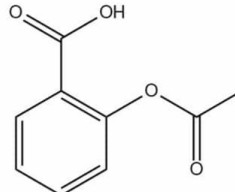

**Input Molecule**

**Original Description:** Aspirin is a widely used analgesic and anti-inflammatory drug that irreversibly inhibits cyclooxygenase (COX) enzymes. It works by acetylating a serine residue in the active site, preventing the synthesis of prostaglandins and thromboxanes.

**User:** Explain how the structural features of this molecule enable its mechanism of action.

**SymForce**
This molecule features an acetyl group (-COCH$_3$) that serves as an irreversible acylating agent for Ser530 in COX-1 and Ser516 in COX-2. The salicylic acid backbone provides proper orientation and binding affinity to the enzyme active site, while the ester linkage is strategically positioned for nucleophilic attack by the serine hydroxyl group...

---

**Mol-LLaMA**
The molecule contains an acetyl group and aromatic ring but lacks specific details about the irreversible binding mechanism or target residue identification...

---

**GPT-4o**
Contains acetylsalicylic acid structure with anti-inflammatory properties, though the explanation of covalent modification remains superficial...

---

**LLaMo**
Shows salicylate structure with acetyl modification but fails to connect structure to specific enzymatic mechanism...

---

**3D-MoLM**
Identifies as NSAID with acetyl group but lacks mechanistic details of COX inhibition...

---

**Mol-Instructions**
Describes as acetylsalicylic acid derivative with general anti-inflammatory effects...

---

810
811
812
813
814
815
816
817
818
819
820
821
822
823
824
825
826
827
828
829
830
831
832
833
834
835
836
837
838
839
840
841
842
843
844
845
846
847
848
849
850
851
852
853
854
855
856
857
858
859
860
861
862
863

Table 7: Case study comparing molecular understanding for morphine analysis.

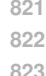

**Input Molecule**

**Original Description:** Morphine is a potent opioid analgesic that binds to $\mu$-opioid receptors in the central nervous system. Its complex polycyclic structure with multiple stereocenters is crucial for high-affinity receptor binding and analgesic efficacy.

**User:** Analyze the stereochemical features that are critical for this molecule's receptor binding.

**SymForce**
This molecule exhibits a rigid phenanthrene-based tetracyclic framework with five stereocenters, particularly the crucial (5R,6S,9R,13S,14R) configuration. The tertiary nitrogen at position 17 provides essential ionic interaction with Asp147 in the $\mu$-opioid receptor, while the phenolic hydroxyl at C-3 forms hydrogen bonds with His297. The $\beta$-oriented hydroxyl at C-6 creates a favorable binding pocket geometry...

**Mol-LLaMA**
Contains multiple ring systems and hydroxyl groups but fails to specify stereochemical requirements or receptor interaction details...

**GPT-4o**
Identifies polycyclic structure with nitrogen-containing rings, though stereochemical analysis is incomplete...

**LLaMo**
Recognizes opioid structure with phenolic groups but lacks detailed stereochemical insights...

**3D-MoLM**
Describes as alkaloid with analgesic properties but provides limited stereochemical analysis...

**Mol-Instructions**
Identifies as opioid compound with basic pharmacological description...

Table 8: Case study comparing molecular understanding for penicillin G analysis.

**Input Molecule**

**Original Description:** Penicillin G is a $\beta$-lactam antibiotic that inhibits bacterial cell wall synthesis by irreversibly acylating transpeptidase enzymes. The strained four-membered $\beta$-lactam ring is essential for its antimicrobial activity.

**User:** Explain how the molecular structure enables antibacterial activity.

**SymForce**
The molecule contains a highly strained $\beta$-lactam ring (azetidin-2-one) with 90° bond angles creating significant ring strain energy ( 27 kcal/mol) that drives nucleophilic attack by Ser403 in DD-transpeptidase. The fused thiazolidine ring provides structural rigidity and proper orientation, while the benzyl side chain fits into the enzyme's hydrophobic pocket. This covalent acylation permanently inactivates the enzyme critical for peptidoglycan cross-linking...

---

**Mol-LLaMA**
Contains $\beta$-lactam structure and sulfur-containing ring but lacks understanding of ring strain and covalent mechanism...

---

**GPT-4o**
Identifies four-membered lactam ring with antibiotic properties, though mechanistic details are superficial...

---

**LLaMo**
Recognizes penicillin core structure but fails to explain ring strain significance or transpeptidase interaction...

---

**3D-MoLM**
Describes as $\beta$-lactam antibiotic with general cell wall inhibition but lacks detailed molecular mechanism...

---

**Mol-Instructions**
Identifies as antibiotic compound with basic structural features...

---

Table 9: Case study comparing molecular understanding for cholesterol analysis.

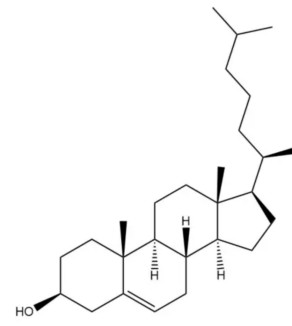

**Input Molecule**

**Original Description:** Cholesterol is a sterol molecule essential for membrane fluidity regulation and serves as a precursor for steroid hormones. Its rigid steroid backbone and polar hydroxyl group enable specific membrane organization.

**User:** Describe how this molecule's structure affects membrane properties.

**SymForce**
This molecule features a rigid steroid backbone (rings A-D) with $\beta$-hydroxyl at C-3 providing amphiphilic character for membrane insertion. The planar steroid rings (thickness 5.2 Å) intercalate between fatty acid chains, reducing membrane fluidity at physiological temperatures. The $\beta$-OH group hydrogen bonds with phospholipid head groups, while the flexible alkyl tail (C-17 side chain) aligns with fatty acid chains. This creates liquid-ordered phase domains critical for membrane organization...

**Mol-LLaMA**
Contains steroid structure with hydroxyl group but lacks understanding of membrane mechanics and phase behavior...

**GPT-4o**
Identifies sterol with multiple rings, though membrane interaction details are incomplete...

**LLaMo**
Recognizes cholesterol structure but fails to explain amphiphilic properties or fluidity effects...

**3D-MoLM**
Describes as membrane component with general structural role but lacks detailed biophysical insights...

**Mol-Instructions**
Identifies as steroid molecule with basic membrane function...

Table 10: Case study comparing molecular understanding for dopamine analysis.

**Input Molecule**

**Original Description:** Dopamine is a catecholamine neurotransmitter that binds to dopamine receptors in the brain. Its catechol ring and ethylamine side chain are essential for receptor recognition and neurotransmission.

**User:** Explain the structural requirements for dopamine receptor binding.

**SymForce**
This molecule contains a catechol moiety (ortho-dihydroxybenzene) where both hydroxyls are essential for hydrogen bonding with Ser199 and Ser203 in D2 receptors. The meta-hydroxyl pattern (3,4-diOH) is crucial as para-hydroxyl analogs show dramatically reduced affinity. The ethylamine chain provides optimal spacing for ionic interaction between the protonated amino group and Asp114. The aromatic ring $\pi$-system enables $\pi$-$\pi$ stacking with Phe390, creating the bioactive conformation necessary for GPCR activation...

**Mol-LLaMA**
Contains aromatic ring and amine group but lacks specificity about hydroxyl positioning and receptor interactions...

**GPT-4o**
Identifies catecholamine structure with neurotransmitter function, though binding specificity is unclear...

**LLaMo**
Recognizes dopamine structure but fails to explain hydroxyl pattern importance or chain length optimization...

**3D-MoLM**
Describes as neurotransmitter with general brain function but lacks detailed binding analysis...

**Mol-Instructions**
Identifies as catecholamine compound with basic neurological role...

Table 11: Case study comparing molecular understanding for vitamin C analysis.

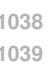

**Input Molecule**

**Original Description:** Vitamin C (ascorbic acid) is an essential antioxidant that prevents scurvy and supports collagen synthesis. Its enediol structure enables electron donation and radical scavenging.

**User:** Analyze the structural features responsible for antioxidant activity.

**SymForce**
This molecule features an enediol system (C2-C3 double bond with adjacent hydroxyls) that enables facile electron donation with low oxidation potential (+0.28 V). The lactone ring constrains the enediol geometry for optimal orbital overlap and resonance stabilization. Upon oxidation, the resulting dehydroascorbic acid maintains biological activity through reversible reduction. The C-6 primary alcohol enhances water solubility while the two-electron oxidation mechanism allows efficient radical termination without forming harmful intermediates...

**Mol-LLaMA**
Contains hydroxyl groups and ring structure but lacks understanding of enediol chemistry and oxidation mechanism...

**GPT-4o**
Identifies ascorbic acid structure with antioxidant properties, though mechanistic details are superficial...

**LLaMo**
Recognizes vitamin structure but fails to explain electron donation capability or radical chemistry...

**3D-MoLM**
Describes as antioxidant vitamin with general health benefits but lacks detailed chemical mechanism...

**Mol-Instructions**
Identifies as ascorbic acid derivative with basic antioxidant function...

Table 12: Case study comparing molecular understanding for taxol analysis.

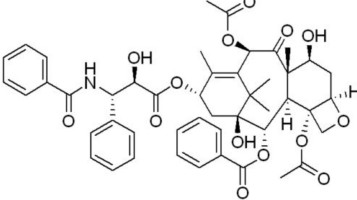

**Input Molecule**

**Original Description:** Taxol (paclitaxel) is a potent anticancer drug that stabilizes microtubules by binding to $\beta$-tubulin. Its complex diterpene structure with multiple chiral centers is crucial for tubulin binding and cytotoxic activity.

**User:** Explain how this molecule's structure enables microtubule stabilization.

**SymForce**
This molecule contains a taxane diterpene core with critical C-2 benzoyl and C-13 side chain that fit precisely into the $\beta$-tubulin binding pocket. The C-13 N-phenylisoserine side chain forms key hydrogen bonds with His229, Asp226, and Arg369 residues. The rigid tricyclic framework maintains proper spatial orientation while the C-2 benzoyl group engages in $\pi$-$\pi$ interactions with Phe272. This binding allosterically stabilizes the straight tubulin conformation, preventing GTP hydrolysis-induced depolymerization and causing mitotic arrest at metaphase...

**Mol-LLaMA**
Contains complex ring systems and multiple side chains but lacks understanding of tubulin interactions and stabilization mechanism...

**GPT-4o**
Identifies anticancer compound with microtubule effects, though binding details are incomplete...

**LLaMo**
Recognizes taxane structure but fails to explain specific protein interactions or stabilization chemistry...

**3D-MoLM**
Describes as chemotherapy agent with general cytotoxic effects but lacks detailed molecular mechanism...

**Mol-Instructions**
Identifies as anticancer drug with basic therapeutic description...

Table 13: Case study comparing molecular understanding for ibuprofen analysis.

**Input Molecule**

**Original Description:** Ibuprofen is a propionic acid derivative that provides anti-inflammatory and analgesic effects through selective COX inhibition. Its chiral center and bulky substituents influence enzyme selectivity.

**User:** Analyze the structural features that determine COX selectivity.

**SymForce**
This molecule contains a $\alpha$-methyl carboxylic acid with (S)-configuration as the active enantiomer that chelates with Arg120 and Tyr355 in COX active sites. The bulky isobutylphenyl group creates steric hindrance in the COX-2 active site (Ile523) compared to COX-1 (Val523), providing modest COX-2 selectivity. Unlike aspirin, ibuprofen acts as a reversible competitive inhibitor through non-covalent binding, with the propionic acid moiety mimicking arachidonic acid substrate...

**Mol-LLaMA**
Contains carboxylic acid and aromatic substituents but lacks understanding of chiral specificity and COX selectivity...

**GPT-4o**
Identifies NSAID structure with anti-inflammatory activity, though selectivity mechanism is unclear...

**LLaMo**
Recognizes propionic acid derivative but fails to explain stereochemical requirements or enzyme interactions...

**3D-MoLM**
Describes as pain reliever with general anti-inflammatory effects but lacks detailed selectivity analysis...

**Mol-Instructions**
Identifies as anti-inflammatory compound with basic therapeutic effects...

Table 14: Case study comparing molecular understanding for retinol analysis.

**Input Molecule**

**Original Description:** Retinol (vitamin A) is essential for vision, cell differentiation, and development. Its polyene structure and terminal alcohol group enable photochemical processes and gene regulation.

**User:** Describe how this molecule's structure enables its biological functions.

**SymForce**
This molecule features a conjugated polyene system with five double bonds enabling photoisomerization from 11-cis to all-trans configuration in rhodopsin. The $\beta$-ionone ring provides conformational rigidity while the terminal primary alcohol enables oxidation to retinal for visual transduction. The extended $\pi$-conjugation system absorbs 500 nm light matching peak sensitivity of rod cells. As retinoic acid, the carboxylic acid derivative binds RAR/RXR nuclear receptors to regulate HOX genes critical for embryonic development...

**Mol-LLaMA**
Contains vitamin structure with alcohol group but lacks understanding of photochemistry and isomerization...

**GPT-4o**
Identifies vitamin A structure with vision functions, though molecular mechanisms are superficial...

**LLaMo**
Recognizes retinoid structure but fails to explain conjugation importance or gene regulation...

**3D-MoLM**
Describes as essential vitamin with general developmental role but lacks detailed photochemical analysis...

**Mol-Instructions**
Identifies as vitamin A compound with basic biological functions...

Table 15: Case study comparing molecular understanding for glucose analysis.

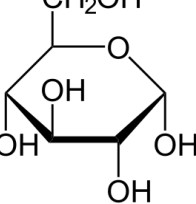

**Input Molecule**

**Original Description:** Glucose is the primary energy source for cellular metabolism and exists in equilibrium between linear and cyclic forms. Its stereochemistry and anomeric center are crucial for enzymatic recognition and transport.

**User:** Explain the structural features important for glucose metabolism and recognition.

**SymForce**
This molecule adopts a $\beta$-D-glucopyranose chair conformation with equatorial hydroxyls minimizing steric interactions. The C-1 anomeric center exists in $\alpha/\beta$ equilibrium (36:64) enabling specific enzyme recognition by glucokinase and glucose transporters...

**Mol-LLaMA**
Contains sugar structure with multiple hydroxyl groups but lacks understanding of conformational equilibria and anomeric effects...

**GPT-4o**
Identifies hexose sugar with energy metabolism role, though stereochemical details are incomplete...

**LLaMo**
Recognizes glucose structure but fails to explain chair conformation or transporter recognition...

**3D-MoLM**
Describes as metabolic substrate with general energy functions but lacks detailed structural analysis...

**Mol-Instructions**
Identifies as carbohydrate molecule with basic metabolic role...

