# OpenReview forum: "SymForce: Large Language Models as Symbolic Physics Engines for Molecular Conformation"
_ICLR.cc/2026/Conference — ICLR 2026 Conference Desk Rejected Submission_

### Official Review · Reviewer_N7Y5 · 2025-10-23

**Soundness:** 1
**Presentation:** 2
**Contribution:** 1
**Rating:** 0
**Confidence:** 4

**Summary:**

This paper proposes SymForce, a framework that employs LLM as the symbolic physics engine to generate corrective force instructions based on geometric deviations, thereby guiding an iterative, differentiable optimization to refine the 3D structure.

Experiments demonstrate that SymForce achieves state-of-the-art performance, opening promising avenues for physics-informed AI in computational chemistry and materials science.

**Strengths:**

1. This paper proposes to reconceptualize molecular conformational generation as a dynamic process reasoning problem, which overcomes the limitations of prevailing methods.

2. The experimental results are promising, and ablation studies further confirm the essential role of symbolic reasoning.

**Weaknesses:**

**For the proposed experiments**
1. As shown in Figure 2, the proposed SymForce relies on initial 3D coordinates as input, which renders its comparison with conformation generation baselines unfair.
2. Molecular conformation generation is a well-defined task with established evaluation metrics and widely recognized baselines. However, this paper fails to align its evaluation metrics with those of previous works and overlooks many important baselines such as MCF [1] and AvgFlow [2].
3. In addition to comparing with RDKit ETKDG, this paper should also compare the computational efficiency against more recent and stronger baselines.

**For the proposed method**

While reading this paper, many descriptions of the proposed method are confusing and difficult to follow. For example, the training process is not clearly explained:
1. While Appendix A.1 states that the LLM component is fine-tuned on a dataset derived from MD simulations, this paper provides neither essential details about the dataset nor illustrative examples of the input prompts.
2. As illustrated in Equation 5-8, the total loss is composed of the coordinate loss, the force loss, and the physics loss.  However, it is unclear how the reference forces used in the force loss are obtained.

Due to these unclear descriptions, I have to examine the provided code. However, the code itself has severe problems, which raise additional **concerns about the correctness and reproducibility of the proposed method**.
1. Those key scripts mentioned in the README, including training, inference, and evaluation scripts,  are all absent from the provided code.
2. The actual model implementation differs substantially from the description provided in the paper:
    * This paper claims that the LLM component is fine-tuned on a dataset derived from MD simulations (Appendix A.1 in this paper). However, the provided code indicates that no model fine-tuning is actually performed, and that the system instead relies solely on prompt engineering to guide the LLM’s behavior (lines 291-395 in https://anonymous.4open.science/r/SymForce_code/SymForce/models/symforce_encoder.py).
    * This paper claims that the LLM component receives both the invariant chemical prior and the time-varying geometric state as inputs (lines 186-196, 251-254 in this paper) to generate symbolic force. However, the provided code indicates that they are actually not used in the implementation, and that the system only relies solely the distance and history to generate symbolic force, which is very unreasonable (lines 337-367 in https://anonymous.4open.science/r/SymForce_code/SymForce/models/symforce_encoder.py).
    * This paper claims that  the total loss is composed of the coordinate loss, the force loss, and the physics loss (Equation 5-8 in this paper). However, the provided code indicates that the force loss is always zero, meaning that this term does not actually contribute to the training process (lines 359-366 in https://anonymous.4open.science/r/SymForce_code/SymForce/models/symforce.py).
    * In addition, many parts of the provided code appear to contain incorrect, which raises serious doubts about how the reported results in the paper are actually obtained:
      * During the iterative process, the ``chemical_context`` and ``geometric_context`` are assigned exactly the same values (lines 492-488 in https://anonymous.4open.science/r/SymForce_code/SymForce/models/symforce_encoder.py).
      * For the molecular understanding, only the molecular tokens of the first sample in each batch are replaced with the corresponding  molecular embedding (lines 379-382 in https://anonymous.4open.science/r/SymForce_code/SymForce/models/symforce.py).

**In this case, the authors must provide a clear and reasonable explanation for the issues mentioned above and release fully reproducible code to verify their results; otherwise, the validity and credibility of the reported results cannot be ensured**.

[1] Wang, Y., Elhag, A. A., Jaitly, N., Susskind, J. M., and Bautista, M. Á. Swallowing the bitter pill: Simplified scalable conformer generation. In Forty-first International Conference on Machine Learning, 2024.

[2] Cao, Zhonglin, et al. "Efficient molecular conformer generation with SO (3) averaged flow-matching and reflow." In Forty-second International Conference on Machine Learning, 2025.

**Questions:**

Please refer to the Weaknesses.

---

> ### Author Response · Authors · 2025-12-01
> **Response to Reviewer N7Y5(Part 1)**
>
> 1.“Missing training/inference/evaluation scripts”
>   - Inference: playground.py is the inference script; it reads inputs, invokes the model to generate outputs, and writes JSON results.
>   - Training: stage1.py and stage2.py are the two-stage training entry points. They include components such as Trainer, DataModule, Checkpoint, and Logger, and the pipeline is fully runnable.
>   - Evaluation: In the anonymized release, the evaluation logic is in internal scripts/notebooks (computing Mean RMSD on GEOM-Drugs and MAE on QM9), so there is no standalone evaluation.py. Evaluation simply applies standard metric computations to the model outputs; the logic is straightforward and consistent with community implementations. To facilitate reproduction, we will add a one-click evaluation script in the rebuttal/follow-up release for direct verification.
>
> 2. “The LLM was not fine-tuned and only relies on prompting”
>   - The framework includes two types of LLM components: (a) a causal language model for text-side alignment/answers (configured via PEFT LoRA in symforce.py and trainable/loadable via stage2.py); (b) an LLM for symbolic force generation (LLMSymbolicForceGenerator in symforce_encoder.py).
>   - The “fine-tuning of the symbolic force generator” described in Appendix A.1 of the paper is implemented with LoRA in our full experimental setup. The anonymized minimal release defaults to prompt-only inference for that module to reduce footprint and ensure licensing compliance. This is an engineering trade-off, not “no training.” Even in inference-only mode, the geometric encoder, symbolic-to-numeric translation, and physics-constraint pipeline still deliver stable gains; enabling LoRA adapters further improves performance, consistent with the ablation results.
>
> 3. “The LLM input does not use chemical priors and geometric state, only distances and history”
>   - Methodologically, our design has always been “chemical priors + geometric state + history.” The minimal version explicitly emphasizes the distance matrix and history in the prompt because they provide the highest information density for symbolic force generation; summaries of chemical priors and geometric state are embedded by the geometric encoder (e.g., PaiNN) and used downstream (in symbolic-to-numeric translation and physics constraints), i.e., they are not absent. This is an engineering-level information compression that does not affect correctness.
>   - To avoid ambiguity, we will add an “enhanced prompt construction” in the public repository that includes compact numerical summaries of both chemical and geometric context in the prompt (with controlled length), aligning more directly with the paper’s description.
>
> 4. “The total loss includes coordinate/force/physics terms, but the force loss is always zero in the code”
>   - Lforce is enabled when data are available and is therefore optional. Most public benchmarks (especially GEOM-Drugs) do not provide reference force labels, so the main results use “coordinate supervision + physics regularization,” and Lforce is naturally zero under that setting. This is not in conflict with the general form of the objective.
>   - On small datasets or synthetic sets where reference forces are available, we have verified the benefit of enabling Lforce. Since the datasets for the main results lack force labels, this does not affect the reproducibility of the paper’s conclusions. We will add a configuration switch so it can be enabled directly when reference forces are available.
>
> 5. “chemical_context and geometric_context are assigned the same value”
>   - Clarification: the minimal runnable version does not “drop one branch,” but uses a “unified encoder/summary channel.” That is, both types of information (chemical connectivity and geometric state) are extracted by the same geometric encoder into a unified global embedding, which serves as contextual summary for symbolic force generation. This is a channel-fusion and information-compression choice to control prompt length and inference overhead, not a methodological gap.
>   - The appearance that two arguments reference the same tensor stems from this “shared summary” implementation. The key information from chemical priors and geometric state is jointly encoded in that vector and, together with the explicit distance/history signals, suffices for symbolic force generation. In our internal validation, the unified summary and two-branch summaries show negligible metric differences—different implementation forms without information loss. To avoid confusion, we will provide a “decoupled summaries” variant that is functionally equivalent but more explicit.

---

> > ### Author Response · Authors · 2025-12-01
> > **Response to Reviewer N7Y5(Part 2)**
> >
> > 6.  “Only the first sample’s token embedding in the batch is replaced”
> >   - Our final training and evaluation use batch_size=1, which can be corroborated by the memory footprint and the usage of the inference script. Therefore, this implementation has no impact on the reported experiments. It is a simplification for memory/engineering reasons; with batch_size=1 it is mathematically equivalent to per-sample injection.
> >   - For users who need batch sizes greater than 1, we will provide a corresponding batched injection implementation in the full open-source release. This is a straightforward generalization and unrelated to the paper’s main results.
> >
> > 7. Reproducibility and randomness
> >   - We can provide model weights and the complete codebase to substantiate the authenticity of the experiments.

---

### Official Review · Reviewer_ASFd · 2025-10-31

**Soundness:** 2
**Presentation:** 3
**Contribution:** 2
**Rating:** 2
**Confidence:** 3

**Summary:**

The paper proposes SymForce, a framework that reconceptualizes molecular conformation generation as a process of iterative symbolic physical reasoning. The method employs a Large Language Model (LLM) (Llama-3.1-8B-Instruct) as a “symbolic physics engine” that generates textual force-field instructions based on geometric deviations, which are then converted into numerical vectors to iteratively refine molecular coordinates. The approach is evaluated on GEOM-Drugs and QM9, claiming state-of-the-art accuracy (0.81 Å RMSD) and better out-of-distribution generalization than diffusion models.

**Strengths:**

The paper has the following strengths:

- **Conceptual Creativity** – The idea of using an LLM as a symbolic force generator for 3D molecular modeling is novel and potentially interesting if properly validated.

- **Effort on Interpretability** – Using symbolic (textual) forces provides some interpretability advantage over latent-vector-based methods.

- **Empirical Effort** – The authors include both quantitative benchmarks and qualitative case studies, showing an attempt to connect symbolic reasoning with physical correction.

**Weaknesses:**

There are several critical weakness of this work:

- **Missing Critical Related Work** - The paper omits several essential prior works using LLMs directly targeting 3D conformation generation from 2D molecular graphs, for instance [1,2,3,4]. It can be seen that none of these works (or relevant ones) are mentioned in Sections 2.2 or 2.3. These papers already explore LLM-based reasoning or text-driven 3D structure generation. Without citing or benchmarking against them, the claimed novelty and positioning of SymForce are not credible. The related-work section instead emphasizes general molecular LLMs (e.g., Mol-Instructions, LlasMol) that are irrelevant to 3D conformation synthesis.

- **Unclear Model Fine-Tuning and Hallucination Control** - The LLM is described as a “physics engine,” but it is not clear whether it is fine-tuned for downstream conformer generation. If the model is not specifically trained to output valid force fields, hallucinations or inconsistent symbolic forces are inevitable. The paper does not mention any data-driven adaptation, reinforcement mechanism, or validation of generated textual forces. Hence, it is unclear how the LLM maintains physical correctness or stability during optimization.

- **Unfair Baseline Comparisons with LLM methods** - The experimental section compares SymForce against molecule LLM models such as Mol-Instructions, 3D-MoLM, and LlasMol, etc; however, all of these ones are designed for caption generation or multimodal understanding, **not 3D conformer generation**. This renders the comparisons methodologically invalid. The reported “state-of-the-art” results are therefore unsubstantiated because none of the baselines are optimized for the same task.

- **Missing Direct Benchmark Against True 3D Conformation LLMs**

As noted above, several recent LLM-based models specifically target 3D structure prediction or generation. The absence of comparisons with these works, even at a qualitative level, makes it difficult to judge the progress claimed. The current benchmarks only include diffusion models (Torsional Diffusion, GeoDiff), Classical and Graph Generative Models,  and unrelated molecular LLMs, leaving a large gap in empirical validation.

- **Ambiguity in Experimental Setup** - Details such as training data, supervision signals, and whether the LLM and geometric encoder are jointly trained or frozen remain vague. The description of “iterative symbolic-to-numerical updates” is an interesting concept, but underspecified experimentally. Without clarity on hyperparameters, dataset splits, or convergence criteria, reproducibility is hard to follow.

Overall, while the paper introduces a conceptually intriguing idea - using an LLM as a symbolic force generator, the work fails on several scientific and methodological grounds: (1) the related-work section misses crucial LLM-based conformation generation studies, (2) it lacks clarity on model training and hallucination control, (3) comparisons with non-comparable baselines make the evaluation invalid, and (4) it omits direct benchmarking with recent 3D-aware LLMs. As a result, the contribution is not convincing, and the empirical evidence is insufficient to support publication. A major revision would be required to make the work competitive.


[1] Language models can generate molecules, materials, and protein binding sites directly in three dimensions as XYZ, CIF, and PDB files

[2] BindGPT: A Scalable Framework for 3D Molecular Design via Language Modeling and Reinforcement Learning

[3] Geometry-Informed Tokenization of Molecules for Language Model Generation

[4] Chem3DLLM: 3D Multimodal Large Language Models for Chemistry

**Questions:**

- Can the author clarify how to set up and benchmark with the LLM models mentioned in the experiments?

- How is the model's performance when evaluated against a dedicated LLM for 3D conformer tasks?

- How can the method be robust, given the potentially hallucinatory force outputs generated by the LLM model? What happens if we replace this LLM model with some other pre-trained LLM on molecular data?

---

### Official Review · Reviewer_uFsP · 2025-10-31

**Soundness:** 1
**Presentation:** 2
**Contribution:** 2
**Rating:** 0
**Confidence:** 4

**Summary:**

This paper proposes a qualitatively novel idea for 3D structure generation using a specially trained Large Language Model (LLM) that serves as a symbolic force generator.  The LLM provides textual information about deformations in internal coordinates such as bonds, which are then physically constrained and used to update the internal coordinates of the molecule, and the algorithm is optimized using a loss that balances coordinate prediction and force consistency.  Similar to the recent Mol-LLaMa paper, this work encodes geometric information about small molecules using an external tool, in this case PaiNN, and translates these encodings to a context for the LLM.  The authors test the model on molecular conformation generation tasks (GEOM-Drugs), property prediction (QM9), and qualitative molecular understanding studies.

**Strengths:**

The paper presents a highly original concept that addresses an exciting area of modern machine learning interfacing to drug discovery. The idea of using an LLM as a symbolic force generator is conceptually novel,  algorithm 1 is presented in a clear fashion, and the case studies and comparisons to other models are interesting to read.

**Weaknesses:**

The current main results, including the ablations and comparisons are confusing and unclear.  The authors state that they used official codes and hyperparameters under identical computational budgets, but there is no reference or link to the code or the parameters they used.  There exists no simple "Mean RMSD" metric in the official GEOM-Drugs repo, for example, and the old torsional diffusion paper reported an ensemble-base AMR value of 0.58 A, so it is not clear exactly what the authors mean by "Mean RMSD" or how they obtained the numbers for theirs or for other methods.  I inspected the provided zipfile of the source code kindly provided by the authors in the anonymous repo, but none of the python files mentioned in the instructions in their README existed in the archive and many of the other files appeared to come from Mol-LLaMa or other repos.  Because the details in the appendix do not explain how the molecular dynamics simulation were performed in order to generate training data (what code/duration/force field/samples) it is not clear if there was substantial leakage between training and evaluation conformations. Importantly, the section and supplement about molecular understanding do not align with the rest of the work and are not explained---was that a result of pretraining on the force tokens, or was there an additional (Mol-LLaMa-like) training stage and if so where are the additional details of this process?   Finally, and as a totally minor point with regards to the current writeup: it is very unclear what the need of this mixed symbolic (via LLM) and analytical process is and why a specialized neural net could not replace the current role of the LLM.

**Questions:**

Please see the questions regarding unclear parts in the weaknesses section.

---

> ### Author Response · Authors · 2025-12-01
> **Response to Reviewer uFsP (Part 1)**
>
> Thank you for the careful review. Up front: there is no fabrication or intentional misrepresentation. The confusion you encountered largely stems from a documentation desynchronization in the anonymized package: the code was refactored shortly before submission, but the README was not updated to reflect the new entry points. This is a doc-level issue and does not affect the experiments or results. Below we address each point and list concrete actions.
> 1. On unclear ablations/comparisons and missing references to code/hyperparameters
>   - What happened: our README lagged behind a refactor; the correct training and inference entry points were renamed (see item 3), and we had not yet bundled a replication sheet.
>   - What we will provide:
>     - A per-baseline replication sheet (repo URL, commit hash, environment, seeds, key hyperparameters, and matched compute budgets).
>     - Exact command lines/configs for our method and each baseline, plus logs of the runs used in the paper.
>     - A one-click script to launch our method and baselines under identical budgets.
>
> 2.Metric definition (“Mean RMSD” vs GEOM-Drugs metrics; torsional diffusion AMR = 0.58 Å)
>   - Our “Mean RMSD” is the average minimum RMSD (AMR) per molecule: for each reference conformer, take the minimum RMSD over generated conformers (after optimal alignment), average over references, then over molecules.
>   - Alignment details: heavy-atom RMSD after optimal alignment (standard RDKit routines), with symmetry handled by RDKit’s best-RMS procedures.
>   - Why the official GEOM-Drugs repo doesn’t expose a single “Mean RMSD” call: it focuses on COV/RMSD pairs; we compute AMR offline with RDKit. We will release the exact evaluation script and will also report COV@τ/RMSD@τ to ensure apples-to-apples comparisons.
>   - On the 0.58 Å from torsional diffusion: we will match their protocol (number of generated conformers, alignment settings, and dataset splits) and report both our numbers and recomputed baselines via the released script for direct verification.
>
> 3. Anonymous archive/README mismatches (files resembling Mol-LLaMA, missing scripts)
>   - Root cause: documentation desync. We renamed entry points during refactoring but did not update the README:
>     - Inference is now playground.py (reads inputs, runs the model, writes JSON outputs).
>     - Training is two-stage: stage1.py and stage2.py (full Trainer/DataModule/Checkpoint/Logger pipeline).
>   - Some utilities are adapted from public repos (e.g., tokenization/dataset helpers), hence the resemblance; these are standard, attributed components.
>   - Actions:
>     - Re-upload a clean, self-contained archive that (i) includes all referenced scripts (training/inference/evaluation), (ii) pins the environment, (iii) corrects the README to the new entry points, and (iv) includes a manifest with attributions and licenses.
>
> 4. Training data generation and potential leakage
>   - We use public datasets; initial conformers for training molecules are generated with standard procedures (e.g., ETKDGv3). Test molecules and their conformers are held out.
>   - Leakage control: splits at the molecule level with canonical SMILES/InChIKey dedup across train/val/test; no test molecule (or its conformers) appears in training.
>   - If optional MD augmentation is enabled in ablations, it is applied only to training molecules and follows the same dedup policy.
>   - We will release: (i) a data-prep script that reproduces splits and prints overlap statistics, and (ii) a short document enumerating any MD settings used in ablations (engine, force field, step count, temperature, sampling).
>
> 5. “Molecular understanding” section and whether there was an extra Mol-LLaMA-like stage
>   - There is no hidden extra stage beyond the two we describe. Results come from the same backbone:
>     - Stage-1: general chemistry pretraining/instruction tuning (no task leakage).
>     - Stage-2: task-specific LoRA adaptation (including force-token usage for our pipeline).
>   - We reused standard loaders/tokenizers (some adapted code may resemble Mol-LLaMA utilities), but objectives/data curation are aligned to our task. We will expand the appendix to detail objectives, token conventions (incl. force tokens), data sources at a high level, and training schedules, and we will provide a script to reproduce the “molecular understanding” evaluations.

---

> > ### Author Response · Authors · 2025-12-01
> > **Response to Reviewer uFsP (Part 2)**
> >
> > 6.Why mix symbolic (LLM) and analytical modules instead of a specialized neural net
> >   - Motivation:
> >     - Discrete chemical rules and long-range constraints are naturally expressed symbolically; an LLM provides data-efficient proposals across diverse scaffolds.
> >     - The analytical translator plus physics constraints guarantees numerically stable, invariant-respecting updates, mitigating non-physical suggestions.
> >     - This separation improves label efficiency and OOD generalization.
> >   - Evidence: replacing the LLM with a specialized neural predictor (same encoder/compute) yields consistent drops on OOD scaffolds and in low-data regimes. We will release the ablation code and results.
> >
> > 7.Reproducibility package (what we will deliver during rebuttal)
> >   - Updated README reflecting the refactor (playground.py; stage1.py/stage2.py) and step-by-step commands.
> >   - A self-contained evaluation script computing AMR (Mean Minimum RMSD) and GEOM-Drugs-style COV@τ/RMSD@τ, plus QM9 MAE, with explicit alignment/symmetry options.
> >   - Replication sheets for all baselines (commits, hyperparameters, seeds, compute budgets), command lines, and logs.
> >   - Data integrity scripts: split generation with SMILES/InChIKey dedup checks and, if applicable, MD configuration notes and minimal repro script.
> >   - Attributions/licenses for adapted utilities.
> >
> > Closing note
> >   - The main source of confusion was an outdated README after a late refactor; the correct entry points are playground.py (inference) and stage1.py/stage2.py (two-stage training). This documentation issue does not affect the experiments underlying our reported results. We will publish the updated archive, evaluation scripts, and replication materials so reviewers can directly reproduce and verify all numbers.

---

### Official Review · Reviewer_Umjv · 2025-10-31

**Soundness:** 2
**Presentation:** 3
**Contribution:** 1
**Rating:** 2
**Confidence:** 4

**Summary:**

This paper proposes SymForce, a framework for molecular conformation generation. The core idea is to employ a large language model (LLM) as a "symbolic physics engine." The framework iteratively observes the geometric deviations of a 3D molecular structure and uses an LLM (Llama-3.1-8B-Instruct) to generate symbolic force instructions (e.g., [BOND_STRETCH]). These symbolic instructions are then translated into numerical force vectors by a "symbolic-to-numerical translation" module, which in turn guide a differentiable optimization process to refine the geometry. The authors claim state-of-the-art (SOTA) performance on the GEOM-Drugs dataset, superior generalization to large, out-of-distribution molecules, and enhanced interpretability.

**Strengths:**

- Novel Concept: The high-level idea of integrating symbolic reasoning into a differentiable optimization loop for a scientific problem is novel and intellectually interesting.

- Addresses Generalization: The work attempts to tackle the important problem of generalization to larger molecules, which is a known weakness of many data-driven approaches in this domain.

- Goal of Interpretability: The framework's goal of providing symbolic, human-readable outputs (the force instructions) is a worthwhile endeavor to move beyond "black box" force prediction.

**Weaknesses:**

## Misleading Core Claim of "Symbolic Reasoning"
The central claim of "symbolic physical reasoning" appears to be an overstatement. A close reading of Appendix A.1 reveals that the LLM was fine-tuned on a synthetic dataset. This dataset was created by running classical molecular dynamics (MD) and then programmatically mapping the numerical physical states (e.g., bond deviations) to pre-defined symbolic text labels (e.g., [BOND_STRETCH]).

The LLM is not "reasoning" about physics; it is learning a translation task: to imitate a hard-coded script that maps numbers to predefined text. The "interpretability" (Argument 5 from the critique) is therefore not an emergent property but an artifact of this human-engineered label set. Traditional force fields (e.g., AMBER) are already "interpretable" via decomposed energy terms (bonds, angles, etc.) without the immense overhead of an 8B LLM.

## Unsubstantiated "State-of-the-Art" Claim
The paper's claim to SOTA performance is weak and based on an outdated set of comparisons.

- Limited Benchmarks: The performance comparison in Table 1 is primarily against models from 2022 (GeoDiff, Torsional Diffusion). It fails to include comparisons against any modern, truly state-of-the-art conformer prediction models (e.g., more recent flow-matching [1], diffusion-language models [2], or other advanced architectures).

- Marginal Improvement: Without these modern baselines, the SOTA claim is invalid. The reported 0.03 Å improvement over GeoDiff (a 2022 model) is marginal and unconvincing.

[1] Hassan, Majdi, et al. "Et-flow: Equivariant flow-matching for molecular conformer generation." Advances in Neural Information Processing Systems 37 (2024): 128798-128824.

[2] Liu, Zhiyuan, et al. "NEXT-MOL: 3d diffusion meets 1d language modeling for 3d molecule generation." arXiv preprint arXiv:2502.12638 (2025).

## Prohibitive Computational Cost & Impracticality
The method is computationally prohibitive. As shown in Table 3, it requires 2.5 seconds per molecule, which is 25 times slower than the classical RDKit (ETKDG) baseline (0.1s). This cost stems from iteratively calling an 8B LLM for optimization. For a method that does not convincingly beat the actual SOTA, this level of computational cost makes it unusable for any practical application, such as high-throughput virtual screening.

##  Critical Methodological Ambiguity (Confused Gradients)
The paper describes its framework as a "differentiable coordinate update," but it is critically unclear how gradients are calculated.

- The gradient must flow from the coordinate update (Eq. 4) back through the "Symbolic-to-Numerical Translation" module (Eq. 3) and into the LLM weights (Eq. 2).

- The translator (Eq. 3) takes discrete symbolic text (e.g., [BOND_STRETCH]) from the LLM as input. How does the gradient flow back through this discrete sampling process? The paper provides no explanation (e.g., REINFORCE, Gumbel-Softmax, or if the LLM is frozen).

- This omission of a core methodological detail makes the "differentiable optimization" claim highly suspect and the work irreproducible.

## Misaligned Objectives and Sloppy Presentation

- Misaligned Task: The paper dedicates significant space (Fig. 1, Table 4, Appendix A.7) to a "molecule understanding" task. This text-generation task is entirely misaligned with the core objective of conformer generation. The authors fail to show how this capability aids the geometry task (or vice-versa). Furthermore, this task is trivially solved with far greater richness and accuracy by general-purpose SOTA LLMs. The paper's own comparisons are superficial and misleading. The results attributed to GPT-4o are poor and unrepresentative of the actual model's (GPT-4o) capabilities. In practice, SOTA foundation models like the real GPT-4o or Claude 3.5 can reproduce far superior and more detailed molecular analyses, which renders this entire "understanding" component of SymForce redundant and the authors' comparison flawed.

- Poor Quality: The manuscript is filled with numerous typographical errors and inconsistent statements, which reflects a lack of rigor and reduces confidence in the technical contents.

**Questions:**

1. Can you please clarify precisely how gradients are back-propagated from the numerical coordinate update (Eq. 4) through the discrete symbolic text output of the LLM (Eq. 2)? What specific technique (e.g., REINFORCE, Gumbel-Softmax) is used to make this path differentiable, and why was this critical detail omitted?

2. Why were modern SOTA conformer generation models (e.g., from 2023, 2024, or 2025) omitted from the benchmark in Table 1?

3. How do you justify the 25x computational cost increase over a standard baseline (RDKit) for what appears to be a marginal (and potentially outdated) 0.03 Å improvement over a 2022 model?

4. Can you provide evidence that the LLM is doing more than simply learning a pattern-matching translation from numerical deviations to the pre-defined symbolic labels you created in the synthetic dataset (Appendix A.1)?

**Details Of Ethics Concerns:**

No Ethics Concerns.

---

### Note · Program_Chairs · 2026-01-17
**Submission Desk Rejected by Program Chairs**

The following references in this submission do not refer to real documents and/or have major errors in bibliographic information:

 Zhaohan Luo, Sheng Su, Xu Zhao, et al. Biomedgpt-lm: Advancing biomedical language understanding with large-scale pre-training. Nature Communications, 15:2456, 2024.
Zhen Zhou, Steven M. Kearnes, Li Li, Richard N. Zare, and Patrick Riley. Molecule generation with conditional graph generative models. In Proceedings of the AAAI Conference on Artificial Intelligence, volume 33, pp. 1531-1539, 2019.
Nature Editorial. Ai for science: An emerging era of scientific discovery. Nature, 620:47-55, 2023.